# The Predictive Role of Serum Lipid Levels, p53 and ki-67, According to Molecular Subtypes in Breast Cancer: A Randomized Clinical Study

**DOI:** 10.3390/ijms25073911

**Published:** 2024-03-31

**Authors:** Ionut Flaviu Faur, Amadeus Dobrescu, Ioana Adelina Clim, Paul Pasca, Catalin Prodan-Barbulescu, Cristi Tarta, Andreea-Adriana Neamtu, Dan Brebu, Carmen Neamtu, Mihai Rosu, Ciprian Duta, Andreea Clim, Gabriel Lazar, Bogdan Totolici

**Affiliations:** 1IInd Surgery Clinic, Timisoara Emergency County Hospital, 300723 Timisoara, Romania; flaviu.faur@umft.ro (I.F.F.); drpascapaul@gmail.com (P.P.); tarta.cristi@umft.ro (C.T.); dr.brebudan@gmail.com (D.B.); duta.ciprian@umft.ro (C.D.); 2X Department of General Surgery, “Victor Babes” University of Medicine and Pharmacy Timisoara, 300041 Timisoara, Romania; 3Multidisciplinary Doctoral School, “Vasile Goldiș” Western University of Arad, 310025 Arad, Romania; 4IInd Obstetrics and Gynecology Clinic “Dominic Stanca”, 400124 Cluj-Napoca, Romania; clim.adelina@yahoo.com; 5Faculty of Medicine, “Victor Babes” University of Medicine and Pharmacy Timisoara, 300041 Timisoara, Romania; catalin.prodan-barbulescu@umft.ro; 6Department I-Discipline of Anatomy and Embryology, Faculty of Medicine, “Victor Babes” University of Medicine and Pharmacy Timisoara, 300041 Timisoara, Romania; 7Faculty of Pharmacy, “Victor Babes” University of Medicine and Pharmacy Timisoara, Eftimie Murgu Sq., Nr. 2, 300041 Timișoara, Romania; aneamtu94@gmail.com; 8Pathology Department, Clinical County Emergency Hospital of Arad, Andrenyi Karoly Str, Nr. 2-4, 310037 Arad, Romania; 9Ist Clinic of General Surgery, Arad County Emergency Clinical Hospital, 310158 Arad, Romania; neamtu.carmen@uvvg.ro (C.N.); mihai.rosu@yahoo.com (M.R.); totolici.bogdan@uvvg.ro (B.T.); 10Department of General Surgery, Faculty of Medicine, “Vasile Goldiș” Western University of Arad, 310025 Arad, Romania; 11Department of Morpho-Functional Sciences II, Discipline of Physiology, “Grigore T. Popa” University of Medicine and Pharmacy, 700115 Iasi, Romania; clim.andreea@umfiasi.ro; 12Department of Oncology Surgery, “Iuliu Hatieganu” University of Medicine and Pharmacy, 400347 Cluj-Napoca, Romania; dr.lazar.gabriel@gmail.com; 13Ist Clinic of Oncological Surgery, Oncological Institute “Prof. Dr. I Chiricuta”, 400015 Cluj-Napoca, Romania

**Keywords:** molecular subtype, HDL-C, neoadjuvant therapy, KI-67, BRCA, biomarker, p53

## Abstract

Dyslipidemia is a component of metabolic syndrome, having an important role in the carcinogenesis of different tumor types, such as prostate, ovarian, or renal cancer. The number of studies on the predictive potential of the different components of the lipid profile with a predictive potential in breast cancer is quite low. The evaluation of the lipid profile was carried out for the 142 patients who benefited from neoadjuvant therapy (NAC) in order to identify a potential predictive biomarker. The serological sample collection was performed sequentially according to a standardized protocol, pre-NAC, post-NAC and 6 months post-NAC after a 6-h pre-collection fast. We also investigated in the general group the presence or absence of the p53 mutation (TP53) and of the mitotic index ki-67, respectively, in relation to the molecular subtypes. The menopausal status, tumor size, family history, grading, Ki-67, p53 and LN metastases have a predictive nature regarding overall survival (OS) (*p* < 0.05), while for disease free survival (DFS), only tumor size, tumor grading, Ki-67 > 14, and p53+ are of predictive nature. The genetic and molecular analysis carried out in our group indicates that 71.67% have a Ki-67 score higher than 14%, and 39% of the patients have the positive P53 mutation. The multivariate analysis in the case of patients included in the TNBC subtype showed that the increased tumor volume (*p* = 0.002) and increased level of HDL (*p* = 0.004) represent predictive factors for the tumor response rate to NAC. High HDL-C levels before NAC and increased LDL-C levels after NAC were associated with the better treatment response in ER-positive and HER2+ breast cancer patients. Increased HDL-C values and tumor volume represent predictive factors as to the response rate to NAC in the case of patients included in the TNBC subtype. Regarding the ER+ and HER2+ subtypes, increased levels of HDL-C pre-NAC and increased levels of LDL-C post-NAC were associated with a better therapeutic response rate. Tumor grading, Ki-67, p53, and LN metastases have a predictive nature for OS, while tumor size, tumor grading, and Ki-67 > 14, and p53+ are predictive for DFS.

## 1. Introduction

Breast cancer represents a worldwide health problem, being the most common form of malignancy in women, ranking second in terms of mortality. In the period 2021–2022, 281,550 new cases of breast cancer were reported, which represents approximately 30% of newly detected cancer cases among women worldwide [1,2,3]. Numerous potentially predictive factors for breast cancer are described in the literature, such as the histological type, histological grade, lymphovascular invasion, tumor size, axillary lymph node status, and hormone receptors. All these factors are currently classified as the gold standard in breast cancer staging [4]. There are biomarkers with prognostic and predictive potential, such as the lipid profile (HDL, LDL, VLDL, Triglycerides, Total Cholesterol, Apolipoprotein A1, Apolipoprotein B), p53, and ki-67, which should be included in routine investigations and which, not infrequently, can influence therapeutic conduct [5,6,7,8]. Dyslipidemia is a component of metabolic syndrome, having an important role in the carcinogenesis of different tumor types, such as prostate, ovarian, or renal cancer [9]. 

The number of studies on the predictive potential of the different components of the lipid profile with a predictive potential in breast cancer is quite low [10,11]. The studies identified in the literature provide heterogeneous data regarding the predictive impact of the lipid profile in breast cancer in relation to molecular subtypes and neoadjuvant therapy [12,13,14,15]. Relatively sparse studies have evaluated the relationship between novel lipid biomarkers, such as apolipoprotein A-I (Apo A-I) and apolipoprotein B-100 (Apo B-100), and cancer risk Apo A-I, the major protein components of HDL cholesterol, and Apo B-100, the major protein components of low-density lipoprotein (LDL) cholesterol, and antiatherogenic and proatherogenic effect manifestation of the HDL and LDL cholesterol, respectively. The analysis of potential biomarkers in breast cancer has been a highly debated topic in recent years, however, with heterogeneous results in the literature [16]. The relation of these potential biomarkers with the molecular subtypes of breast cancer represents an important objective, and once some predictive directives are certified, we could develop new therapeutic guidelines. Another important biomarker in carcinogenesis is represented by P53 or TP53. The alteration of p53 has been investigated with particular interest in recent years and studies have shown that the p53 gene represents the most frequent mutation in carcinogenesis processes [17]. The frequency of these mutations in the case of breast carcinoma is quite heterogeneous in relation to molecular subtypes. The mutation of the *P53* gene may represent an early event in tumor progress because it is obvious in the in situ phase of cancer growth. Additionally, the *P53* mutation probably stimulates cell proliferation and renders the phenotype aggressive. The availability of detecting mutant p53 protein on formalin-fixed paraffin-embedded (FFPE) tissue has allowed retrospective studies of patients with a long follow-up. Ki-67 represents another important parameter in tumor carcinogenesis, being a non-histone nuclear protein that correlates with cell growth. The Ki-67 expression varies through cell cycle, with different expression levels in the G1, G2/M, and S phases, but undetectable in the G0 phase. Ki-67 associates with cell cycle progress and the short half-life confers it as an effective biomarker for assessing the growth fraction of tumor cells [18,19,20,21].

## 2. Results

We have evaluated 313 cases of BC followed by surgical resection between 2020 and 2022, in the Ist Clinic of Oncological Surgery, Oncological Institute “Prof Dr I Chiricuta” Cluj Napoca. We applied the inclusion criteria for our study and defined a study batch of 233 cases. Of the 233 cases, 142 benefited from neoadjuvant therapy (NAC), and the remaining 91 cases benefited from primary surgical therapy. The stratification of the study group was made according to the St. Gallen classification; therefore, 40.77% (n = 95 cases) were included in the luminal subtype A, 24.89% (n = 58 cases) in the luminal type B, 6% (n = 14 cases) in the HER2+ subtype, and 28.32% (n = 66 cases) in the TNBC subtype. A total of 21% of the patients enrolled in the study had a family history of malignancy and approximately 70% of them had a BMI over 25 kg/m^2^. From a histological point of view, ductal carcinoma had an incidence of approximately 88.02%, followed by lobular carcinoma with a percentage of 11.15%.

Among the patients who benefited from neoadjuvant therapy (n = 142 cases), 88.02% (n = 125 cases) had between 4 and 8 cycles of neoadjuvant therapy, while 11.97% (n = 17 cases) had less than 4 cycles of therapy. Regarding the response rate to NAC according to the Response Evaluation Criteria in Solid Tumors (RECIST-Miller-Payne Classification) guidelines, version 1.1, 64.78% presented a partial response (PR) or complete response (pCR) to NAC therapy (Table 1).

If we refer to the tumor size, the test carried out shows that 36.48% (n = 85) of the patients were staged as cT2, and 33.9% (n = 79) were staged as cT3. Regarding tumor grading, 41.63% (n = 97) were moderately differentiated G2 tumors, followed by poorly differentiated G3 tumors in approximately 29.61% (n = 69) of the cases, while 28.75% (n = 67) were classified as well-differentiated G1 tumors.

The analysis of the genetic and molecular profile of the study group reveals that 71.67% (n = 167 cases) have a Ki-67 above 14%, while 39% (n = 91 cases) of the patients present a positive P53 mutation. Among the patients with positive P53 mutation (n = 91), 15 (16.48%) were included in the luminal A subtype, 9 (9.89%) in the luminal B subtype, 10 (10.98%) in the HER2+ subtype, and 57 patients (62.63%) in the TNBC subtype. Another eloquent aspect of our evaluation is related to the presence of the BRCA1/2 mutation, which was reported as positive in 18.8% (n = 44 cases) of the cases, thus confirming the hereditability of the tumor. Genomic DNA was purified from EDTA-anticoagulated blood using the QiaSymphony instrument (Qiagen, Hilden, Germany). Genotyping of a panel of 20 pathogenic *BRCA1* and 10 pathogenic *BRCA2* variants was carried out using TaqMan Low-Density Arrays on the ABI 9700 instrument (Applied Biosystems, Foster City, CA, USA) as recommended by the manufacturer.

There are five simple cancer-based criteria for BRCA: (1) ovarian cancer; (2) breast cancer diagnosed when patients are 45 years or younger; (3) two primary breast cancers, both diagnosed when patients are 60 years or younger; (4) triple-negative breast cancer; and (5) male breast cancer. A sixth criterion—breast cancer plus a parent, sibling, or child with any of the other criteria—can be added to address family history. Criteria 1 through 5 are considered the MCG criteria, and criteria 1 through 6 are considered the MCGplus criteria. In the literature, approximately 20% of breast neoplasias have a hereditary nature, the BRCA1/2 genes having a cardinal role in their transmission.

Regarding neoadjuvant therapy regimens, 40.14% of the cases benefited from Carboplatin therapy, while 59.85% benefited from Anthracycline/Taxane therapy (Table 2). Analyzing the mutant status of p53 at the level of the general group (n = 233) it was found that 91 patients (39.05%) presented a positive mutation. Referring to the molecular subtype according to the St Gallen classification, 61 cases (67.03%) with a p53 mutation were included in the TNBC subtype (See Table 3), 11 cases (12.08%) were classified as HER2+, 10 cases (10.98%) as Luminal B, and 9 cases as Luminal A (9.89%).

We analyzed the p53 status at the level of the study group and the possibility of its association with the tumor response rate to the NAC therapy (See Figure 1 and Table 4). In the case of patients included in the TNBC subtype (n = 61), the mutant status of p53 was identified in 49 cases, while the wildtype was identified in 12 cases. In the case of TNBC patients, no impact of the p53 status on the complete tumor response rate was identified (*p* = 0.797). A statistically significant correlation was found in the case of TNBC patients, classified as cT1-2, having neoadjuvant therapy with Carboplatin; they presented pCR in 57.3% of cases (*p* < 0.00001). In the case of patients belonging to the Luminal A (n = 9), Luminal B (n = 10), and HER2+ (n = 11) subtypes, we recorded a p53 mutant status in 13 cases, and a p53 wildtype status in 17 cases, without a statistically significant impact of the p53 status on the tumor response rate to NAC (*p* = 0.325). The hormonal status is an important aspect in terms of tumor response rate, therefore in the case of patients with negative hormonal status, significant therapeutic responses were recorded (*p* = 0.0001). 

We compared the average values of the lipid parameters in the group that benefited from neoadjuvant therapy (n = 142) and we obtained statistically significant values regarding the TG and LDL-C values, which presented a significant increase post-NAC compared to the time of diagnosis (before -NAC) and which maintained their increased values in the dynamics 6 months post-NAC *(p* < 0.05).

According to a multivariate analysis (see Table 5), the larger tumor size (*p* = 0.011), high HDL-C level (*p* = 0.005), and LDL-C (*p* = 0.003) were independent predictive factors of the efficacy of NAC. A high HDL-C level before NAC and increased LDL-C level after NAC were associated with better treatment response in ER-positive and HER2+ breast cancer patients. 

A study published by Wei Tian et al. on the analysis of the lipid profile in the case of patients who benefit from neoadjuvant therapy concludes increased values of TC, TG, and LDL-C, but without analyzing them based on the molecular subtype of the tumor. In 2022, Xinru Wang et al. published an analysis including 220 patients regarding the association between the histological type of the tumor and lipid profile, concluding the association between the HDL-C and LDL-C values and the absence of positive hormonal status (PR, ER), thus the low values of HDL-C and LD-C being associated with the ER- and PR- subtypes [22,23].

The multivariate analysis in the case of patients included in the TNBC subtype showed that the increased tumor volume (*p* = 0.002) and increased level of HDL (*p* = 0.004) represent predictive factors for the tumor response rate to NAC. In the case of LDL-C values, we did not obtain statistically significant data regarding tumor response to therapy (See Table 6 and Table 7).

The TC levels were slightly increased before the last cycles of chemotherapy compared to the prechemotherapy levels in all three regimen subgroups. 6 months after chemotherapy, TC levels returned to baseline levels (0.145 [0.084, 0.234]; z = 5.72; *p* < 0.05). The TG levels increased significantly during chemotherapy in all chemotherapy regimen groups. 6 months after chemotherapy completion, TG levels were slightly decreased but were still higher than the baseline levels (prechemotherapy level) in all groups. No significant difference in changes in the TG levels was noticed between the groups. The LDL-C levels were significantly increased during chemotherapy (0.137 [0.071, 0.198]; z = 5.71; *p* < 0.05), but were restored to baseline levels 6 months after chemotherapy completion. Significantly lower LDL-C levels were noticed 6 months after chemotherapy completion compared to the baseline values in the Carboplatin group (0.127 [0.088, 0.218]; z = 5.82; *p* < 0.05). The HDL-C levels decreased significantly during chemotherapy in all regimen groups, and 6 months after chemotherapy completion the levels were nearly restored to baseline levels. The levels of APO-A and APO-B showed minimal fluctuations post-NAC and 6 months after the completion of NAC (*p* > 0.05) regardless of the type of chemotherapy undertaken. A study carried out by Youzhao Ma et al. reveals increased values of triglycerides (TG), total cholesterol (TC), and LDL-C, and low values of HDL-C after NAC (*p* < 0.001) [24].

In order to evaluate the predictive nature of p53 (positive) and Ki-67 (increased values > 14) of the disease-free interval (DFS) and long-term survival (OS-overall survival) we applied a univariate and multivariate analysis in the general group; the analysis included the following variables: age, presence or absence of menopause, family history of the patients, tumor size, tumor grading, Ki-67, lymphatic metastases (LN metastases), and p53.

The univariate and multivariate analysis reveals that age has a predictive nature regarding OS (*p* = 0.001, 0.117 [0.071, 0.212]; z = 5.71;) but not for DFS (0.127 [0.061, 0.162]; z = 5.58; *p* < 0.05). The menopausal status, tumor size (0.145 [0.081, 0.182]; z = 5.30; *p* < 0.05), family history, grading, Ki-67, p53, and LN metastasis have a predictive nature regarding OS (0.177 [0.081, 0.212]; z = 5.11; *p* < 0.05), while for DFS, only tumor size, tumor grading, Ki-67>14, and p53+ are of predictive nature (0.187 [0.094, 0.262]; z = 5.92; *p* < 0.05) (Table 8/Figure 2).

In the logistic regression, tumor size, lymph node metastases, and p53 positivity were associated with recurrence with a more prominent predictive effect (*p* < 0.05). P53/Ki-67 levels must be divided into categorical variables (positive vs. negative, high index vs. low index, and negative) in our logistic regression and Cox regression models (Table 9).

## 3. Discussions

Our study performs an exhaustive analysis of genetic and molecular parameters of a predictive nature regarding the response rate to the NAC therapy. The literature presents absolute heterogeneity regarding the existence of biomarkers with predictive capacity in the case of breast cancer patients who benefit from neoadjuvant therapy. The genetic and molecular analysis carried out in our group indicates that 71.67% have a Ki-67 score higher than 14%, and 39% of the patients have the positive P53 mutation. Among the patients with positive P53 mutation, 16.48% were included in the luminal A subtype, 9.89% in the luminal B subtype, 10.98% in the HER2+ subtype, and 57 patients, i.e., 62.63%, in the TNBC subtype. 

A special category is represented by the patients included in the TNBC subtype; the mutant status of p53 was recorded in 49 cases, and the wildtype in 12 cases. In the case of TNBC patients, no impact of the p53 status on the complete tumor response rate (*p* = 0.797) was identified. In the case of patients belonging to the Luminal A, Luminal B, and HER2+ subtypes, we recorded a p53 mutant status in 13 cases, and a p53 wildtype status in 17 cases without a statistically significant impact on the tumor response rate to NAC (*p* = 0.325). 

Bertheau reported that TP53 might be predictive for chemotherapy response in a particular setting: In a pooled analysis of 144 breast cancers from three series, this group found that pCR rates in TP53 mutant tumors strongly depended on the type of chemotherapy, and were significantly higher after high-dose cyclophosphamide (36%) as opposed to standard-dose (4%) or no cyclophosphamide (12%). The effect was most pronounced in ER-negative tumors, where the pCR rate after high-dose (not lower dose) cyclophosphamide among TP53 mutant cases reached 71% [25].

The menopausal status, tumor size, family history, grading, Ki-67, p53, and LN metastases have a predictive nature regarding OS (*p* < 0.05), while for DFS, only tumor size, tumor grading, Ki-67 > 14, and p53+ are of predictive nature. Pan Y, Yuan Y, Liu G, Wei Y (2017) conducted a study on the immunohistochemical analysis of p53 and Ki-67 in the case of patients with TNBC, which revealed their possible stratification according to the level of tumor aggressiveness and integrated Ki-67 and p53 as potential prognostic biomarkers.

In a recent report, Carey et al. described that in the CALGB 40601 trial, the p53 signature was independently associated with high pCR rates in 305 patients with HER2-positive breast cancer. There are further previous studies on the predictive effect of TP53 status in breast cancer that were summarized in a meta-analysis by Chen in 2012 [26]. Investigating 26 studies comprising 3.476 cases, the authors concluded that TP53 aberrations were associated with a higher response to neoadjuvant chemotherapy, particularly for anthracycline-based regimes [27]. 

The multivariate analysis in the case of patients included in the TNBC subtype showed that the increased tumor volume (*p* = 0.002) and increased level of HDL (*p* = 0.004) represent predictive factors for the tumor response rate to NAC. High HDL-C levels before NAC and increased LDL-C levels after NAC were associated with a better treatment response in ER-positive and HER2+ breast cancer patients. In the literature, there are heterogeneous results regarding the impact of the lipid profile on the tumor response rate to NAC according to the tumor subtype. Xinru Wang et al. published a study comparing the lipid profile between 2 study subgroups that include 70 ER (estrogen receptor) negative patients and 73 PR (progesterone receptor) negative patients, respectively [28,29]. In the case of negative PR patients, increased values of LDL-C and lipoprotein (LP) levels were revealed compared to the positive PR patients. In the case of negative PR patients, increased values of LDL-C and LP were recorded compared to positive PR patients (*p* < 0.05). Another study published in 2020 by Sung Mi Jung et al. underlines the impact of lipid profile values in the breast tumor carcinogenesis process [30]. The TC and TG levels are generally increased in breast cancer patients, and increased levels of LDL-C and VLDL-C and low levels of HDL-C are associated with the tumor development process. Fanli Qu et al. published a retrospective study enrolling 533 patients with breast tumors who benefited from NAC therapy [31]. During the study, the analysis of the lipid profile (TC, TG, HDL-C, LDL-C) was performed before and after the NAC therapy. High HDL-C levels before NAC and increased LDL-C levels after NAC were associated with a better treatment response in ER-positive breast cancer patients. 

## 4. Materials and Methods

We have evaluated 313 cases of BC followed by surgical resection between 2020 and 2022, in the Ist Clinic of Oncological Surgery, Oncological Institute “Prof Dr I Chiricuta” Cluj Napoca. For inclusion in the study, standard criteria were established: (1) age over 18 years (2) female sex (3) no history of malignancy (4) complete oncological file without therapeutic gap (5) no background hypolipidemic therapy (6) pretherapeutic lipid profile analysis. We applied the inclusion criteria for our study and defined a study batch of 233 cases. Of the 233 cases, 142 benefited from neoadjuvant therapy (NAC), and the remaining 91 cases benefited from primary surgical therapy. All cases were histopathologically and immunohistochemically confirmed by core biopsy before therapy. Four-micrometer-thick tissue sections from the surgical specimens fixed in 10% formalin and embedded in paraffin were reviewed, and representative tissue blocks were selected. The evaluation of the lipid profile was carried out for the 142 patients who benefited from neoadjuvant therapy (NAC) in order to identify a potential predictive biomarker. The serological sample collection was performed sequentially according to a standardized protocol, pre-NAC, post-NAC, and 6 months post-NAC after a 6-h pre-collection fast. From the point of view of lipid biomarkers, we used total cholesterol (TC), triglycerides (Tg), high-density lipoprotein cholesterol (HDL-C), low-density lipoprotein cholesterol (LDL-C), and apolipoproteins A and B. For the evaluation of the response to neoadjuvant therapy, we used a standardized protocol: (1) We performed pre-NAC tumor marking using the harpoon thread (2) The evaluation of the response to neoadjuvant therapy was performed on clinical-imaging bases (standard breast ultrasound and MRI). According to the Response Evaluation Criteria in Solid Tumors (RECIST- Miller-Payne Grading) guidelines, version 1.1, a reduction in the sum of the diameters of the target lesions ≥30% was classified as a clinical partial response (PR) [32,33,34,35]. An increase in the sum of the diameters of the target lesions ≥20% was considered clinically progressive disease (PD). Tumors that did not shrink sufficiently to qualify for a PR, and did not increase sufficiently to qualify for PD were classified as clinically stable disease (SD). The absence of any residual tumor lesions in any excised breast tissue or lymph nodes was defined as a pathological complete response (pCR). We also investigated in the general group the presence or absence of the p53 mutation (TP53) and of the mitotic index ki-67, respectively, in relation to the molecular subtypes. Patients were systematically followed up 3 months, 6 months, 1 year, and 2 years after surgery. The aim of this study was to establish potential predictive biomarkers in relation to tumor molecular subtypes. 

### 4.1. Immunohistochemistry Analysis and Molecular Typing

The hormonal status of the patients (ER-estrogen receptor and PR-progesterone receptor), and the HER2 status and Ki 67 index, respectively, were evaluated to outline a molecular stratification according to the St. Gallen classification. The hormonal expression of both ER and PR was considered positive if a percentage of over 1% was obtained. If HER2 was evaluated as 3+ by immunohistochemical staining or recorded over 2.0-fold growth by fluorescence in situ hybridization, the HER2 expression was considered positive. The Ki67 value was defined as the proportion of positively stained cells (500–1000) among the total number of cancer cells in the invasive front of the tumor [36].

The interpretation of the histopathological, immunohistochemical, and molecular results was carried out by two anatomopathologists independently. We evaluated the TP53 status in relation to tumor molecular subtypes.

The histopathological quality control was performed prior to DNA isolation. Only core biopsies with an invasive tumor area = 20% were eligible. DNA was isolated from formalin-fixed and paraffin-embedded tissue (FFPE) by automated DNA extraction via QIAsymphony. p53 was evaluated in our molecular group (TNBC and HER2+), for which a tissue microarray (TMA) constructed from pre-operative punch biopsies was available. A mouse monoclonal antibody directed against p53 protein was used in 1:50 dilution on a Ventana Benchmark autostainer (Ventana, Tucson, AZ, USA) [37].

### 4.2. Statistical Considerations

Statistical analysis was performed using SPSS 24.0 for Windows (SPSS, Inc., Chicago, IL, USA) which was used for data analysis. The different types of associations between molecular parameters and the lipid profile were examined using χ^2^ tests. The multivariable analysis of pCR, overall survival (OS), and disease-free survival (DFS) was carried out using a binary logistic regression model. Normally distributed continuous data were expressed as means (SD) and were assessed using the analysis of variance (ANOVA), independent-sample *t*-test, or paired *t*-test. Nonparametric data were analyzed using the Mann–Whitney and Wilcoxon tests. Two-sided tests were performed to declare statistical significance at *p* < 0.05.

### 4.3. Ethical Consent

All processes approached during the study with the inclusion of human subjects benefited from the approval of the ethics commission according to national and international standards in direct relation to the Helsinki Declaration of 1964. This article does not include studies on laboratory animals. The consent mentioned above was received from, and approved by, each participant in the study (The Ethics Commission for Research and Development Activities and for Quality Assurance of Clinical Trials of the “Prof Dr Ion Chiricuță” Oncological Institute in Cluj Napoca, appointed by the manager’s decision (IOCN no. 189-03.06.2021-Application no. 10442).

## 5. Conclusions

In conclusion, we can state that increased HDL-C values and tumor volume represent predictive factors as to the response rate to NAC in the case of patients included in the TNBC subtype. Regarding the ER+ and HER2+ subtypes, increased levels of HDL-C pre-NAC and increased levels of LDL-C post-NAC were associated with a better therapeutic response rate. Tumor grading, Ki-67, p53, and LN metastases have a predictive nature for OS, while tumor size, tumor grading, Ki-67 > 14, and p53+ are predictive for DFS.

## Figures and Tables

**Figure 1 ijms-25-03911-f001:**
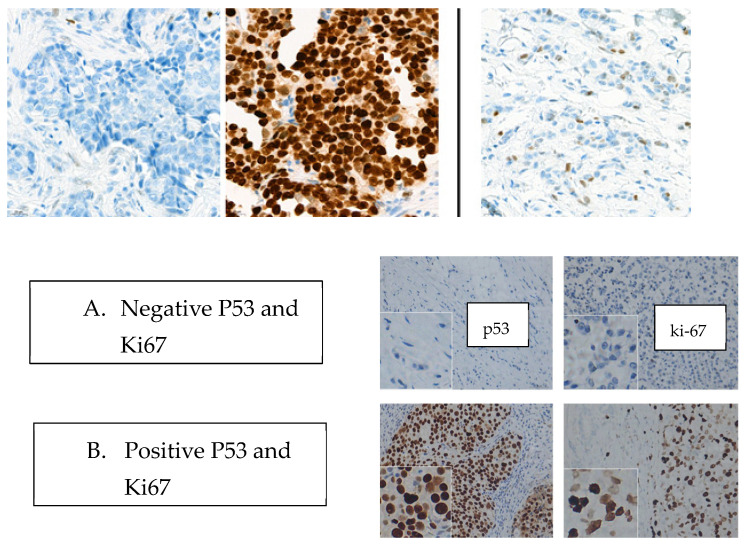
Microscopic aspects regarding the pattern of p53 and Ki-67. Mutant pattern 0% and over 60% Wildtype pattern 1–59% (Magnificationx200).

**Figure 2 ijms-25-03911-f002:**
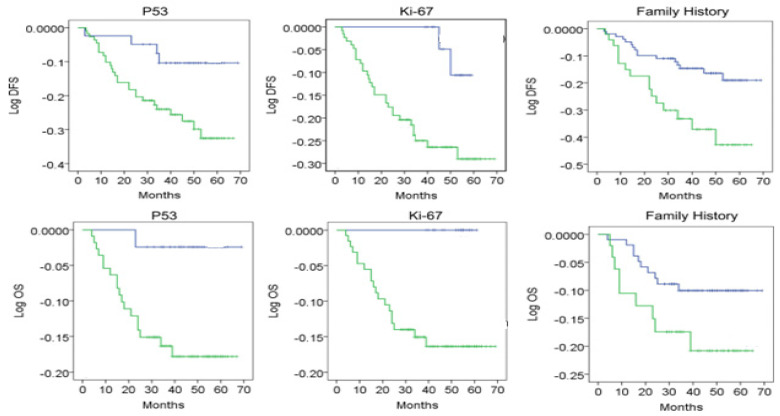
Evaluation of DFS and OS in correlation cu P53, Ki-67, family history. The blue line represents negativ cases with P53 (-)/Ki-67<14%/Family history (-) and the green line represents positive cases with P53 (+)/Ki-67>14%/Family history (+).

**Table 1 ijms-25-03911-t001:** General characteristics of the study group.

Characteristics	n	%
**Molecular Subtype (St Gallen Classification)**		
Luminal A	94	40.77
Luminal B	58	24.89
Her2+	14	6
TNBC	66	28.32
**Age**		
Over 50 years	125	53.64
Under 50 years	108	46.36
**Family history**		
Yes	49	21
No	184	78.96
**Menopause**		
Yes	103	44.2
No	130	55.7
**BMI (kg/m^2^)**		
Under 25 kg/m^2^	66	28.32
25–29.9 kg/m^2^	95	40.77
Over 30 kg/m^2^	72	30.9
**Histological type**		
Ductal	193	82.8
Lobular	26	11.15
Medullar	14	6
**Chemotherapy cycles (n = 142)**		
Under 4	17	11.97
4–8	125	88.02
**cN**		
Positive	147	63.09
Negative	86	36.90
**cT**		
T1	29	12.44
T2	85	36.48
T3	79	33.9
T4	40	17.16
**Grading**		
G1	67	28.75
G2	97	41.63
G3	69	29.61
**Ki-67**		
Over 14%	167	71.67
Under 14%	66	28.32
**Post-NAC response**		
Yes (pCR+PR)	92	64.78
No (SD+PD)	50	35.21

**Table 2 ijms-25-03911-t002:** Genetic and molecular characteristics of the study group.

*Characteristics*	n	%
** *Ki-67* **	233	
*>14%*	167	71.67
*<14%*	66	28.32
** *P53* **		
*Negative*	142	60.94
*Positive*	91	39.05
** *BRCA 1/2 mutation* **		
*Positive*	44	18.8
*Negative*	183	81.11
** *Chemotherapy regimens* **	142	
*Carboplatin*	57	40.14
*Anthracyclines/Taxanes*	85	59.85

**Table 3 ijms-25-03911-t003:** The association between p53 status and pCR according to molecular subtypes.

Molecular Subtype					
*TNBC* (*n* = 61)	n	% pCR	OR	95% CI	*p*
**P53 status**					
Wildtype	12	39.5	1	-	0.797
Mutated	49	45.3	1.02	0.056–1.85	
**Histological type**					
Ductal	55	46.2	1	1	0.567
Lobular/others	6	0.0	n.a.	n.a.	
**Tumor grade**					
G1-2	47	32.2	1	-	0.854
G3-4	14	46.1	n.a.	0.91–3.15	
**Clinical tumor status (cT)**					
cT1-2	39	44.21	1	-	0.001
cT3-T4	22	21.3	0.24	0.12–0.95	
**Type of chemotherapy**					
Carboplatin	37	57.3	1	-	<0.00001
Anthracyclines/Taxanes	24	34.3	2.77	1.74–4.98	
** *HER2+ (n=11)+Luminal A (n=9)+Luminal B (n=10)* **					
**P53 status**					
Wildtype	17	34.3	1	-	0.325
Mutated	13	28.1	0.74	0.40–1.25	
**ER/PR status**					
ER-/PR-	11	44.2	1	-	0.0001
ER+ and/or PR+	19	21.8	0.29	0.21–0.59	
**Histological type**					
Ductal	28	30.5	1	-	0.559
Lobular/others	2	49.2	0.44	0.03–6.55	
**Tumor grade**					
G1-2	18	26.3	1	-	0.225
G3-4	12	34.6	1.25	0.62–2.22	
**Clinical tumor status (cT)**					
cT1-T2	17	29.7	1	-	0.895
cT3-T4	13	33.5	1.11	0.51–1.74	
**Type of chemotherapy**					
Carboplatin	18	27.6	1	-	0.226
Anthracyclines/Taxanes	12	33.5	0.69	0.40–1.29	

**Table 4 ijms-25-03911-t004:** Dynamics of the lipid profile in correlation with the therapeutic moment (Pre-NAC, Post-NAC and 6 m after NAC).

Neoadjuvant Chemotherapy Group		Pre-Chemotherapy (Pre-NAC)	Post-Neoadjuvant Chemotherapy (Post-NAC)	6 m after Chemotherapy
**TC**	Mean ± SD	4.98 ± 1.08	5.41 ± 1.12	5.12 ± 1.01
**TG**	Mean ± SD	1.38 ± 0.71	1.98 ± 0.91 *	1.88 ± 0.81 *
**LDL-C**	Mean ± SD	2.77 ± 0.77	3.22 ± 0.67 *	2.92 ± 0.88
**HDL-C**	Mean ± SD	1.34 ± 0.34	1.19 ± 0.29	1.33 ± 0.34
**Apo-A**	Mean ± SD	1.82 ± 0.29	1.64 ± 0.89	1.91 ± 0.31
**Apo-B**	Mean ± SD	0.89 ± 0.21	1.21 ± 0.34	1.38 ± 0.21

***** Significant correlation.

**Table 5 ijms-25-03911-t005:** Multivariate analysis of baseline characteristics according to the response to NAC (Luminal A, Luminal B, and HER2+).

Characteristics	Clinical Response		
	Hazard Ratio	95%CI	*p*-Value
**Age**	**0.396**	**0.186–1.035**	**0.121**
**Menopause**	**0.958**	**0.403–1.963**	**0.911**
**Ki-67 expression**	**0.745**	**0.456–1.245**	**0.206**
**Tumor size**	**0.523**	**0.285–0.788**	**0.011**
**TG**	**0.662**	**0.356–1.23**	**0.123**
**HDL**	**0.478**	**0.125–0.844**	**0.005**
**LDL**	**0.356**	**0.278–0.856**	**0.003**

**Table 6 ijms-25-03911-t006:** Multivariate analysis of baseline characteristics according to the response to NAC in TNBC.

*Characteristics*	Clinical Response		
	Hazard Ratio	95%CI	p-Value
** *Age* **	**0.316**	**0.183–0.535**	**0.121**
** *Menopause* **	**0.858**	**0.403–1.163**	**0.921**
** *Ki-67 expression* **	**0.645**	**0.457–0.945**	**0.156**
** *Tumor size* **	**0.543**	**0.275–0.988**	**0.002**
** *TG* **	**0.762**	**0.316–0.923**	**0.117**
** *HDL* **	**0.578**	**0.135–0.822**	**0.004**
** *LDL* **	**0.254**	**0.278–0.455**	**0.631**

**Table 7 ijms-25-03911-t007:** Dynamic analysis of the lipid profile depending on the type of chemotherapy used.

Parameter	Regimen	Pre-NAC	Post-NAC	6 m after NAC
	Mean ± SD	Mean ± SD	Mean ± SD	Mean ± SD
TC	Carboplatin	4.96 ± 0.89	**5.23 ± 1.32**	**4.95 ± 0.91**
	Anthracycline/Taxane	**5.12 ± 1.09**	**5.20 ± 1.01**	**5.01 ± 0.98**
TG	Carboplatin	**1.77 ± 0.77**	**2.17 ± 1.23 ***	**2.15 ± 1.07 ***
	Anthracycline/Taxane	**1.78 ± 0.65**	**2.23 ± 1.17 ***	**1.92 ± 1.12 ***
LDL-C	Carboplatin	**2.78 ± 0.89**	**3.04 ± 0.98**	**2.71 ± 0.55**
	Anthracycline/Taxane	**2.71 ± 0.76**	**3.02 ± 0.82 ***	**2.77 ± 0.77**
HDL-C	Carboplatin	**1.29 ± 0.56**	**1.26 ± 0.45**	**1.27 ± 0.25**
	Anthracycline/Taxane	**1.33 ± 0.23**	**1.11 ± 0.24**	**1.33 ± 0.34**
APO-A	Carboplatin	**1.82 ± 0.21**	**1.83 ± 0.45**	**1.80 ± 0.34**
	Anthracycline/Taxane	**1.72 ± 0.34**	**1.74 ± 0.54**	**1.73 ± 0.22**
APO-B	Carboplatin	**0.89 ± 0.23**	**0.92 ± 0.22**	**0.91 ± 0.34**
	Anthracycline/Taxane	**0.79 ± 0.33**	**0.82 ± 0.23**	**0.80 ± 0.43**

***** Significant correlation.

**Table 8 ijms-25-03911-t008:** Univariate and multivariate survival analysis for disease-free survival (DFS) and overall survival (OS).

	Univariate		Multivariate	
Parameters	*p*	HR (95% CI)	*p*	HR (95% CI)
**Age (>50 vs. <50)**	0.598	1.21 (0.27 ± 3.15)	0.001	6.10 (1.24 ± 20.86)
**Menopause (yes vs. no)**	0.655	1.22 (0.45 ± 3.31)	0.054	0.21 (0.08 ± 1.06)
**Family history (yes vs. no)**	0.741	0.71 (0.094 ± 5.31)	0.022	3.89 (1.56 ± 9.51)
**Tumor size (T1-T4)**	0.001	5.16 (2.28 ± 8.18)	0.001	6.13 (1.61 ± 15.48)
**Tumor grade**	0.031	1.269 (1.05 ± 5.93)	0.012	1.57 (0.91 ± 5.35)
**KI-67 (>14 vs. <14)**	0.037	29.52 (0.179 ± 3275)	0.045	1,191,123 (0.00 ± 2.117 × 10^200^)
**P53 (+ vs. −)**	0.051	6.28 (0.97 ± 51.44)	0.005	16.70 (2.13 ± 144.03)
**LN metastasis**	0.001	7.13 (2.41 ± 24.21)	0.003	6.14 (1.73 ± 18.99)

**Table 9 ijms-25-03911-t009:** Logistic regression analysis of factors predicting recurrence.

Parameters	B	S.E.	Wald	df	Sig.	Exp (B)	95% CI
**Age (>50)**	0.334	0.423	0.339	1	0.499	1.23	0.456–4.125
**Menopause (+)**	−0.114	0.451	0.052	1	0.798	0.845	0.322–2.225
**Histological type**			0.154	2	0.922		
**Ductal**	0.328	1.326	0.019	2	0.897	1.192	0.123–14.127
**Lobular**	−0.165	0.966	0.124	1	0.785	0.718	0.112–4.756
**Medullar**	0.163	0.987	0.112	1	0.744	0.777	0.125–12.124
**Tumor size**			5.645	2	0.058		
**T1-T2**	0.856	0.655	2.541	1	0.115	2.325	0.814–6.123
**T3-T4**	1.322	0.877	5.442	1	0.018	6.235	1.326–27.123
**LN metastasis (+)**	1.655	0.455	11.245	1	0.001	4.741	1.921–11.547
**Family history (+)**	0.877	0.651	3.745	1	0.051	2.385	0.992–5.744
**Ki-67 > 14%**	1.493	0.835	2.370	1	0.111	3.747	0.722–18.746
**P53 (+)**	1.455	0.654	5.101	1	0.021	4.218	1.220–14.345
**Constant**	−4.296	0.829	28.021	1	0.000	0.012	

## Data Availability

Data are contained within the article.

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
