# Peer review of "The Predictive Role of Serum Lipid Levels, p53 and ki-67, According to Molecular Subtypes in Breast Cancer: A Randomized Clinical Study"

_ijms, 2024, doi:10.3390/ijms25073911_

Round 1
Reviewer 1 Report
Comments and Suggestions for Authors
In this manuscript, Faur et al. analyzed the genetic and molecular parameters of a predictive nature regarding the response rate to NAC therapy. The methods were thoroughly described, and the findings are interesting. Minor aspects should be addressed:
1. In the abstract and introduction, please define DFS and OS before using the abbreviations.
2. In Section 2, Line 96-99, please double-check the case number and the percentage. Currently, there are 232 cases instead of 233, and one of the percentages does not seem correct.
3. Table 2, it is not clear how the BRCA1/2 mutation positive and negative are calculated.
4. The authors should mention how the ki-67 cut-off of 14% was selected.
Author Response
Response to Reviewer 1 Comments
- In the abstract and introduction, please define DFS and OS before using the abbreviations.
We made the modification and define DFS and OS - overall survival (OS) (p < 0.05), while for disease free survival (DFS),
- In Section 2, Line 96-99, please double-check the case number and the percentage. Currently, there are 232 cases instead of 233, and one of the percentages does not seem correct.
We made the modification - Of the 233 cases, 142 benefited from neoadjuvant therapy (NAC), and the remaining 91 cases benefited from primary surgical therapy. The stratification of the study group was made according to the St. Gallen classification, therefore 40.77% (n=95 cases) were included in the luminal subtype A, 24.89% (n=58 cases) in the luminal type B, 6% (n=14 cases) in the HER2+ subtype, and 28.32% (n=66 cases) in the TNBC subtype.
- Table 2, it is not clear how the BRCA1/2 mutation positive and negative are calculated.
Genomic DNA was purified from EDTA-anticoagulated blood using the QiaSymphony instrument (Qiagen, Hilden, Germany). Genotyping of a panel of 20 pathogenic BRCA1 and 10 pathogenic BRCA2 variants were carried out using TaqMan Low-Density Arrays on the ABI 9700 instrument (Applied Biosystems, Foster City, CA, USA) as recommended by the manufacturer.
There are 5 simple cancer-based criteria for BRCA : (1) ovarian cancer; (2) breast cancer diagnosed when patients are 45 years or younger; (3) 2 primary breast cancers, both diagnosed when patients are 60 years or younger; (4) triple-negative breast cancer; and (5) male breast cancer. A sixth criterion-breast cancer plus a parent, sibling, or child with any of the other criteria-can be added to address family history. Criteria 1 through 5 are considered the MCG criteria, and criteria 1 through 6 are considered the MCGplus criteria.
- The authors should mention how the ki-67 cut-off of 14% was selected
The decision regarding the cut-off value of KI-67 was made arbitrary on the following research in specialized literature where there is no consensus regarding this cut-off value.
Thank you for your support!
Reviewer 2 Report
Comments and Suggestions for Authors
In this manuscript, the authors investigated the role of serum lipids, some key genes like p53 and ki67 in the prediction of breast cancers. Ki-67, p53 and LN metastases have a predictive nature regarding OS and tumor size for the DFS. Multivariate analysis showed that increased tumor volume and increased level of HDL represent predictive factors for the tumor response rate to NAC. In the ER+ and HER2+ subtypes, increased levels of HDL-C pre-NAC and increased levels of LDL-C post-NAC were associated with a better therapeutic response rate and so on.
The predictive role of serum lipid profiles in breast cancer has been previously investigated by other researchers (Doi: 10.4048/jbc.2020.23.e32), potentially decreasing the novelty of this manuscript. Nevertheless, the author has also incorporated a combination of Ki67, p53, and other factors for prediction purposes.
Some comments on this manuscript:
1. For the Immunohistochemistry Analysis/TMA, representative images showing the different grades of marker expression are not present in this manuscript. Please include figures illustrating the varying levels of expression for these markers.
2. Utilize Kaplan-Meier survival curves to illustrate the differences in overall survival/DFS among different groups.
3. Potential prognostic factors associated with OS were assessed through univariate and multivariate Cox analysis. LASSO regression analysis could be utilized to construct a prognostic model incorporating different identified factors. The construction and validation of this prognostic model will be depicted through AUC curves.
4. Develop and validate a nomogram designed to predict the overall survival of patients with breast cancer.
5. Please carefully review for typographical and grammatical errors throughout the manuscript.
Comments on the Quality of English Languagenone
Author Response
Please see the attachement

Round 2
Reviewer 2 Report
Comments and Suggestions for Authors
Thank you for the author's revision. Please label the scale bar and p value in the figures and survival curve. From my standpoint, I have no additional questions.
Comments on the Quality of English LanguageNone
